# Connecting Diagnostics and Clinical Relevance of the α-Gal Syndrome—Individual Sensitization Patterns of Patients with Suspected α-Gal-Associated Allergy

**DOI:** 10.3390/nu17091541

**Published:** 2025-04-30

**Authors:** Uta Jappe, Tahmina Kolaly, Mareike S. de Vries, Askin Gülsen, Arne Homann

**Affiliations:** 1Division of Clinical and Molecular Allergology, Priority Research Area Chronic Lung Diseases, Research Center Borstel, Leibniz Lung Center, Airway Research Center North (ARCN), Member of the German Center for Lung Research (DZL), 23845 Borstel, Germanyaskingulsen@hotmail.com (A.G.); a.homann@gmx.com (A.H.); 2Interdisciplinary Allergy Outpatient Clinic, Department of Pneumology, UKSH Campus Lübeck, University of Lübeck, 23538 Lübeck, Germany; 3Division of Cardiology, Pulmonary Diseases, Vascular Medicine, University Hospital Duesseldorf, 40225 Düsseldorf, Germany

**Keywords:** allergy diagnostics, biomarker, sensitization pattern, α-Gal, glycan, cross-reactive carbohydrate determinants, anaphylaxis, singleplex assay, multiplex assay

## Abstract

Background/Objectives: Sensitization to the carbohydrate antigen α-Gal is associated with allergic reactions against different types of food that contain α-Gal (e.g., mammalian meat). This form of allergy is termed α-Gal syndrome (AGS), and the diagnosis is challenging due to delayed symptom onset and cross-reactivity with multiple mammalian products. It is estimated that AGS is underdiagnosed, pointing to an unmet need for patient care. Methods: Sera from patients with suspected AGS based on clinical history were analyzed by ImmunoCAP and the IgE cross-reactivity immune profiling (ICRIP) system specifically developed by us. IgE from patient sera against different forms of α-Gal was analyzed using α-Gal-containing analytes and negative controls. Results: Sera from 33 patients with suspected AGS were analyzed. Sera from 22 patients yielded a clearly positive signal (>0.35 kU/L) for IgE against α-Gal in ImmunoCAP. For 7 of the remaining 11 patients with negative or ambiguous (IgE level between 0.1 and 0.35 kU/L) results in ImmunoCAP, ICRIP analyses supported the suspected association of the allergy symptoms with IgE against α-Gal components. This component-resolved analysis helps the allergist to provide an individual diagnosis for each patient. Conclusions: The diagnosis of AGS is challenging. An interplay between clinical history and lab analysis via ImmunoCAP and the specifically developed ICRIP system helps patients and allergists in establishing the correct diagnosis, thereby preventing accidental exposure and recurrent AGS episodes.

## 1. Introduction

In 2008, it was discovered that IgE associated with meat allergy is not only directed against peptides (e.g., bovine serum albumin (BSA) from cattle, Bos d 6, which before 2008 was considered the major meat allergen together with other mammalian serum albumins (Sus s 1 (pork), Can f 3 (dog), and Fel d 2 (cat) [1]), but also against the glycan epitope galactose-α-1,3-galactose (α-Gal)) [1,2]. It was identified as a significant allergenic epitope via severe anaphylactic reactions to cetuximab, a chimeric oncological treatment antibody. Chimeric molecules consist of both a murine and a human part. As α-Gal is expressed by non-primate mammals, some bacteria, and parasites and is present in the saliva of some tick species [3], but not expressed by humans. It is the N-glycosylation sequence in the murine part of the treatment antibody via which the α-Gal epitope becomes part of the drug [2]. Further associations with anti-α-Gal-IgE have been observed with other approved biologics, e.g., infliximab and natalizumab [4,5]. The significance of this glycan epitope lies in the fact that (A) it has induced anaphylaxis to the biological at first application, pointing to a non-detected pre-existing sensitization [2,4]; that (B) α-Gal was found to be relevant for patients with anaphylaxis to red meat and innards [6,7,8,9]; (C) that this anaphylaxis was delayed to the extent that rendered adequate allergy diagnostics nearly impossible because, at that time, no clinician considered an anaphylaxis reaction time of >6 h possible [9]; and (D) that repetitive bites by certain tick species are now considered to be the most important sensitization route in US and also in European patients, with different tick species being responsible for cases in the US and Europe [3,10]. This was the first time that anti-glycan IgE could definitely be associated with clinically relevant allergic reactions when compared to “classical” cross-reactive carbohydrate determinants (CCDs) that represent less clinically relevant pan-allergens throughout the plant kingdom [11,12].

It has been described that the concentration of the oligosaccharide α-Gal is higher in innards than in muscle meat [7], thus linking clinically relevant meat allergy reactions (A) to the type of meat and (B) to the amount of consumed α-Gal (i.e., the total amount of meat/innards), representing a dose-dependent allergic reaction. In cases where vast amounts of α-Gal-containing meat or innards are consumed, anaphylaxis occurs even after several minutes. The animal tissue consumed—and in general referred to as “meat”—contains muscle meat, tendons, fat, blood (including immunoglobulins), and innards. Several further meat allergens may induce allergic reactions, but α-Gal seems to be the most important and clinically relevant one. The symptom range is characterized by several distinct phenotypes, which is why the term α-Gal syndrome (AGS) was coined [13,14]. The incidence rate of diagnosed AGS is in the range of 3778–8462 cases in the US per year [15,16]. The prevalence of anti-α-Gal-sIgE in the US, studied in a cohort of military recruits, was found to be 6% in total, with rates of up to 39% in Arkansas, correlating with the geographical distribution of the lone star tick *Amblyomma americanum* [17]. Interestingly, the prevalence of α-Gal-sIgE-positive (≥0.10 kU/L) individuals with pre-existing allergies in a German allergy unit was 19.9%, and a patient-reported tick bite increased the likelihood of sIgE against α-Gal [18]. The onset of AGS occurs mainly during adulthood after mammalian meat and meat products have been tolerated without any problems, sometimes even as a first and only allergy manifestation in advanced age. AGS patients often report urticaria and gastrointestinal symptoms. In some patients, hives are absent, but they suffer from itching and flush. Symptoms can involve only one organ system but can occur in combination. Sometimes the bronchial system is additionally affected. The fact that sole gastrointestinal manifestation occurs (endoluminal food allergy) [19] is considered to be a most relevant differential diagnosis to inflammatory bowel disease. Even cardiovascular afflictions have been identified [20]. Thus, meat allergy is a very complex type of allergy as it involves IgE against different types of mammalian meat, e.g., beef and pork (primary meat allergy), sometimes related to milk (mainly driven by the milk allergens Bos d 4, Bos d 5, and Bos d 8), and being α-Gal-associated with increasing incidence [11,13,21,22]. The latter has been described to occur with and without additional gelatin allergy, as well as with and without allergic reactions to milk [13]. Cat IgA (Fel d 5) has been identified as an α-Gal-containing cat allergen [23]. This example already shows the complexity of IgE cross-reactivity associated with glycan and peptide epitopes from different mammalian species. Moreover, the clinical relevance of these cross-reactivities is not entirely clear and is left for individual judgement by the allergologist. Cross-reactivity has also been described for gelatin-containing food, for biologicals (cetuximab, infliximab), some drugs like anti-venoms, gelatin-based plasma expanders and gelatin-containing drug formulations (capsules), vaccines, and all drug preparations that have been produced by mammalian tissues or with the help of mammalian hybridoma cells [6,24]. To date, it has not been elucidated why some AGS-patients react to milk and/or gelatin and most do not and whether distinctive phenotypes are based upon certain immune parameters, like sensitization profiles and/or the ratio of total serum IgE and specific IgE.

In this study, the binding patterns of 33 well-characterized patients with suspected/manifested meat allergy were analyzed via ImmunoCAP and a newly developed multianalyte array, the IgE cross-reactivity immune profiling (ICRIP) [7]. IgE binding to different forms of α-Gal (CTX, HSA-α-Gal, and Bos d TG) was investigated. The summary of clinical symptoms, clinical history, ImmunoCAP, and ICRIP sheds light on the specific challenges of the diagnosis of α-Gal-dependent allergic reactions and the previously described sensitization mechanism via tick bites [3].

## 2. Materials and Methods

### 2.1. Study Group

Patients with suspected α-Gal-dependent meat allergy were recruited during visits to the allergy outpatient clinics of the Medical Clinic Borstel and the Interdisciplinary Allergy Outpatient Clinic, University Hospital Center, Luebeck, Germany, on an ongoing basis. The diagnostic process as outlined in Figure 1 was followed. Inclusion criteria were suspected allergy to meat/innards, anaphylaxis, and severe urticaria due to unknown causes. Patient characteristics are described in Appendix A. Patients with no history for meat allergy, but specific (s)IgE to meat and α-Gal, were included as meat-/α-Gal-sensitized individuals. The patients were characterized by a standardized questionnaire and the sensitization to α-Gal via different meat extracts, whole milk protein, single milk allergens, and gelatin via ImmunoCAP (ImmunoCAP, Phadia AB, Thermo Fisher Scientific, Freiburg, Germany and Uppsala, Sweden; for in vitro results, see Table 1, Table 2 and Appendix A). This study was approved by the Ethics Committee of the University of Lübeck (approval number 11-166 and 13-086). Patients provided their informed consent in writing. IgE-mediated food allergy to meat/innards was diagnosed only if there was also a history of an acute systemic allergic reaction. Food challenge tests were not performed in the course of this study. However, some patients presented with results of oral challenge tests performed outside the study setting, and those data were included in the patients’ characteristics (Appendix A).

### 2.2. Detection of Allergen-Specific IgE Antibodies by ImmunoCAP

Anti-α-Gal-IgE can routinely be determined in the patients’ serum via ImmunoCAP (Phadia AB, Thermo Fisher Scientific, Freiburg, Germany and Uppsala, Sweden) with bovine thyroglobulin (Bos d TG) as the α-Gal-containing allergen. Therefore, all patient sera were analyzed for IgE binding to Bos d TG (Table 1). Furthermore, IgE specific to the following commercially available test allergens were measured in order to characterize the individual sensitization patterns: α-Gal-containing beef, pork, and mutton meat extract, as well as gelatin. Non-α-Gal containing analytes were Bos d 4 (α-lactalbumin), Bos d 5 (β-lactoglobulin), Bos d 6 (bovine serum albumin), and Bos d 8 (casein) (all from Phadia AB, Thermo Fisher Scientific, Freiburg, Germany and Uppsala, Sweden; Table 1 and Table 2). In addition, total serum IgE (normal serum level <100 kU/L) and baseline serum tryptase (normal range < 11.4 µg/L) were determined (Appendix A). Some sera were diluted in order to obtain a sufficient volume for analysis. The assays were performed on ImmunoCAP^®^ 250 (Phadia AB, Thermo Fisher Scientific, Freiburg, Germany and Uppsala, Sweden). The instrument threshold for a positive signal was 0.1 kU/L according to the manufacturer (website https://www.thermofisher.com/phadia/wo/en/our-solutions.html?icid=phadia, most recent access on 26 April 2025). Note that the cut-off value of ≥0.35 kUL is commonly used for allergy diagnostics. Whereas a cut-off value of ≥0.1 kU/L has been shown to confirm the diagnosis of AGS [25], in a more recent publication, the cut-off value of ≥0.35 kU/L and the different ImmunoCAP classes have been shown to be of discriminative value in the phenotyping of AGS and its dynamics [26].

### 2.3. In Vitro Cross-Reactivity Analysis of sIgE Against Different α-Gal-Containing Proteins by sIgE Cross-Reactivity Immune Profiling (ICRIP)

In addition to the ImmunoCAP^®^ singleplex allergy test, the sIgE cross-reactivity immune profiling (ICRIP) based on a dot blot platform previously established by us was applied for serum characterization (Figure 2). For a detailed description of the platform, please see Homann et al. [27]. Using α-Gal-containing analytes as well as negative controls allows for the detection of α-Gal-dependent and -independent IgE binding. Using the same compounds in singleplex and multiplex in vitro tests in ImmunoCAP and ICRIP assays, respectively, we were able to directly perform an inter-individual comparison of sensitization patterns. The ICRIP system allows for a determination of the differential binding patterns of serum IgE from patients with suspected α-Gal-dependent meat allergy, depending on the affinity for specific α-Gal-containing structures. Analytes representing α-Gal are cetuximab (CTX, Erbitux^®^, Merck, Darmstadt, Germany), human serum albumin (HSA) linked to an α-Gal trisaccharide (HSA-α-Gal_3_, Dextra Laboratories, Berkshire, UK), and bovine thyroglobulin (Bos d TG, Sigma Aldrich/Merck, Darmstadt, Germany). As negative controls in terms of glycosylation, the analytes infliximab (IFX, Remicade^®^, MSD Sharp & Dohme GmbH, Munich, Germany), HSA (Sigma Aldrich/Merck, Darmstadt, Germany), and BSA/Bos d 6 (Sigma Aldrich/Merck, Darmstadt, Germany) were included in the assay (Figure 2).

## 3. Results

### 3.1. Demographic and Clinical Characteristics of Patients with Suspected AGS

A total of 33 patients with suspected AGS were recruited during routine visits to the allergy outpatient clinics. A total of 18 male and 15 female individuals were aged between 17 and 78 yrs, with the age distribution being as follows: 2 were ≤20 yrs old, 4 ≤30 yrs, 6 ≤40 yrs, 5 ≤50 yrs, 9 ≤60 yrs, 6 ≤70, and 1 ≤80, indicating that most patients with AGS/suspected AGS are above 40 years (Appendix A). A total of 24 patients (73%) with suspected AGS remembered to have had at least one tick bite, and 5 of these 24 patients reported persistent inflammatory reactions at the injection site (Appendix A). A total of 23 patients (70%) had pre-existing atopic diseases. For 17 patients, risk factors for anaphylaxis could be documented: stress (n = 7), alcohol in combination with meat consumption (n = 7), exercise (n = 5), NSAID (n = 2), menstruation (n = 1), and infection (n = 1). Several distinct patient groups were identified that showed clinical symptoms after the ingestion of meat/innards only (17 patients), meat/innards plus milk (“dairy”, 9 patients), and meat/innards and gelatin (5 patients). Among these patients with suspected AGS, the following common clinical phenotypes could be identified: 23 patients (70%) reacted with skin symptoms combined with symptoms at other organ systems (16 patients had urticaria, and 12 patients (additional) angioedema). Three patients (9%) only showed gastrointestinal symptoms (endoluminal α-Gal allergy). The symptoms are detailed in Figure 3 and Appendix A.

#### 3.1.1. ImmunoCAP Analysis of Patient Sera

Total serum IgE concentrations were normal (<100 kU/L) in 9/33 patients, whereas elevated total serum IgE concentrations were detected in 21/33 patients (Appendix A), and very high concentrations (>100 kU/L) in 3 patients (P05, P12, and P27). Tryptase concentrations were normal in all patients, indicating no underlying mast cell disease. ImmunoCAP analysis with a panel of α-Gal-containing analytes (Bos d TG, milk protein, pork meat, bovine meat, mutton, and gelatin) and α-Gal-negative molecules (Bos d 4, Bos d 5, Bos d 6, and Bos d 8) was performed for all 33 sera from patients with suspected AGS (Appendix A). The sera of 24/33 patients had elevated total serum IgE above 100 kU/L, but without a correlation with IgE concentration against Bos d TG. In the following two sections, we present the results for patients that were positive or negative in the commercial α-Gal ImmunoCAP test with Bos d TG.

#### 3.1.2. ImmunoCAP Analysis of Patient Sera Positive for sIgE Against Bos d TG

ImmunoCAP analyses of patient sera show the most meaningful results for the single allergen Bos d TG. As expected and published previously, robust results were also obtained with the allergen source bovine meat extract (Figure 4A) [7,13,19,28]. IgE against non-α-Gal-containing analytes were much lower in concentration (Figure 4B). Positive ImmunoCAP results (sIgE against α-Gal/Bos d TG) were obtained for 24 sera (Table 1). Two patient sera (P07 and P13) were below the threshold for a positive allergy assessment of 0.35 kU/L and, therefore, were also considered as negative. Of these 22 sera with specific IgE antibodies against α-Gal, 18 sera were clearly associated with sIgE against bovine and pork meat extract and/or milk protein with ImmunoCAP results >0.35 kU/L (Table 1). Three sera positive for sIgE against Bos d TG below the threshold of 0.35 kU/L had only a weak association with meat extract allergens, and no other association with any allergen tested (P10, P26, and P31). The one remaining serum with sIgE against Bos d TG (P08) was negative for all other α-Gal-containing allergens. Only a weak association with milk was detectable, confirming the previously published clinical association (sIgE to milk protein was 0.21 kU/L, Table 1) [29]. In three patient sera, a weak signal for sIgE against gelatin could be detected (>0.1 and <0.35 kU/L, Table 1). Two of these three sera (P21 and P30) showed positive IgE signals for all other α-Gal-containing analytes. Serum from patient P13 showed a broad signal distribution with sIgE against all analytes tested (α-Gal- and non-α-Gal-containing), but all below the threshold of 0.35 kU/L.

#### 3.1.3. ImmunoCAP Analysis of Patient Sera Negative for sIgE Against Bos d TG

Nine sera from patients with suspected AGS were negative in ImmunoCAP test for anti-α-Gal-IgE (Table 2). Additionally, two patient sera (P07 and P13) showed signals below the threshold for a positive allergy assessment of 0.35 kU/L and, thus, were also considered as negative for sIgE against α-Gal. Moreover, in ImmunoCAP tests, the serum from patient P07 was not reactive with any other ImmunoCAP analyte. Serum test from patient P13 resulted in a broad signal distribution for all allergen sources and (single) allergens tested in ImmunoCAP, but all below the threshold of 0.35 kU/L. Four patient sera (P02, P03, P09, and P18) were negative for sIgE against Bos d TG and all other tested allergens. Further sIgE serum signals were each below the threshold of 0.35 kU/L: sIgE against bovine meat (P01 and P25), sIgE against milk protein and Bos d 5 (P05), and IgE against milk protein (P08, published previously [29]). Lastly, serum from patient P29 was positive above 0.35 kU/L for sIgE against milk protein, Bos d 4, Bos d 8, and bovine meat.

#### 3.1.4. Patient Sera IgE Analysis by sIgE-Cross-Reactivity-Immune Profiling (ICRIP)

By ICRIP analysis, all 33 patient sera were tested in this immunoblot-based assay for the assessment of sIgE cross-reactivity to different α-Gal-containing analytes. This way, sera from patients with suspected AGS were characterized in terms of (A) IgE-binding affinity and (B) -sensitivity to different α-Gal-containing analytes. These α-Gal-containing analytes are the therapeutic antibody cetuximab, human serum albumin (HSA) linked to a chemically defined α-Gal structure (HSA-α-Gal_3_), and bovine thyroglobulin (Bos d TG) mirroring the molecule used for ImmunoCAP analysis (Figure 2).

A total of 12 of 33 patients showed an IgE pattern clearly associated with α-Gal-dependent meat allergy with sIgE signals against all α-Gal-containing analytes CTX, HSA-α-Gal_3_, and Bos d TG (Table 3 and Appendix A). Additionally, six patient sera showed a signal for sIgE against CTX only, and five sera showed sIgE binding to Bos d TG only. Moreover, variations in sIgE-binding patterns were detectable (Table 3 and Appendix A): CTX/Bos d TG (P09, P27, P31), CTX/IFX (P10), CTX/IFX/HSA-α-Gal_3_/Bos d TG (P30), and CTX/HSA-α-Gal_3_ (P04). Of the nine sera with negative results for sIgE against α-Gal in the ImmunoCAP system, the ICRIP analysis resulted in positive sIgE signals in five sera against Bos d TG (P01, P02, P03, P18, and P29). P09 was IgE-positive for Bos d TG and CTX. Two sera did not show sIgE binding to any analyte (P05, P15). ICRIP test of one serum (P25) resulted in a signal of sIgE binding to Bos d 6 (BSA), a protein allergen without α-Gal. Two patient sera with an ImmunoCAP signal below 0.35 kU/L were either completely negative in ICRIP (P07) or positive for CTX (and not Bos d TG, P13). Sera from three patients were negative for α-Gal-containing components in ImmunoCAP and ICRIP (P05, P15, and P25; Table 1 and Table 3). The ImmunoCAP result from patient P07 for sIgE against Bos d TG (0.11 kU/L) was a borderline positive result with no sIgE signal for any α-Gal-containing analyte in the ICRIP analysis, and thus, can most likely be considered as negative for relevant α-Gal-specific IgE binding.

## 4. Discussion

The α-Gal syndrome (AGS) shows a broad individual spectrum of clinical symptoms, mostly affecting the skin but with the potential to include all organ systems. As allergy in general is highly individual, AGS patients show diverse symptoms (Figure 3, Appendix A), with the clinical phenotypes of patients pointing to a potential AGS diagnosis. Patients with AGS are in need of a clear diagnosis and subsequent education about α-Gal-containing food and drugs causing anaphylaxis. However, as already known from other forms of allergy, the ImmunoCAP diagnostic system with α-Gal (Bos d TG) alone may not be sufficient for confirming the diagnosis in every case [7,30]. Due to these aspects, the delay in symptom development, and the lack of knowledge on this entity in medical personnel in more general clinical practice, AGS is still underreported, which is why it is of such vast importance to improve the respective diagnostic measures. As it has been shown in case reports [30], ImmunoCAP may not always be sensitive enough; so, a broader in vitro multianalyte assay is of considerable value, particularly because of the lack of authorized skin prick test (SPT) solutions. Additionally, because of the expected increase in (α-Gal sensitizing) tick populations due to the climate change in the near future, an accurate diagnosis of the AGS is of the essence. Interestingly, three patients (P02, P06, and P24) are professional cooks with a regular exposure to mammalian meat and who experienced many tick bites in the past that have been described as the sensitization route for AGS, summarized in [3]. This situation prompts research on the detection of useful and valid laboratory parameters/biomarkers as well as diagnostic systems.

In this study, we used a complementary diagnostic system (ICRIP) that can be applied to either detect anti-α-Gal IgE antibodies not detected by ImmunoCAP or confirm the sensitization detected by ImmunoCAP. Sera from 33 patients with suspected AGS were analyzed by both ImmunoCAP and ICRIP. We recruited patients with AGS and investigated them for clinical characteristics and the routine singleplex IgE measure (ImmunoCAP) as well as their immune profile via a multianalyte assay based on different α-Gal-carrying molecules (ICRIP). The singleplex method ImmunoCAP is a widely used in vitro diagnostic system that quantifies specific IgE antibodies to a variety of allergen sources and single allergens in serum. This system is particularly useful in AGS diagnosis, because it can specifically measure sIgE concentration against the α-Gal-containing molecule Bos d TG. This quantification is crucial since an elevated α-Gal-specific IgE concentration together with characteristic symptoms is a significant biomarker of AGS. In addition, quantification analyses can be helpful to document the dynamics of this entity, for example, in cases where new tick bites have occurred and probably boosted the allergy. One example is patient P33, who experienced a decrease in α-Gal-sIgE under avoidance and an increase in sIgE concentration after a new tick bite (sIgE concentration changed from 2.31 kU/L at the first occurrence of AGS, decreased to 0.24 kU/L during avoidance of exposition for 29 months, and increased again to 5.78 kU/L shortly after a tick bite). Unlike SPT, which may yield variable results—and for which no standardized diagnostic solution has been authorized to date—ImmunoCAP quantifies specific IgE antibodies directly linked to the allergen in question. The delayed nature of AGS makes it difficult to diagnose based on patient history alone, as symptoms may be mistaken for allergic reactions to other food allergen sources or digestive disorders. ImmunoCAP tests can distinguish α-Gal sensitization from other forms of food or environmental allergies by isolating the α-Gal-specific IgE response. This differentiation is valuable because some patients with AGS may not experience reactions to milk or other animal-based products that contain mammalian proteins and α-Gal in varying concentrations, allowing for more tailored dietary advice. Despite its advantages, there are certain limitations and considerations when using ImmunoCAP for AGS diagnosis, as certain individuals may have detectable levels of α-Gal-specific IgE without exhibiting AGS symptoms, which has been shown for hunters experiencing many tick bites but only rarely develop AGS [31]. Therefore, elevated sIgE alone cannot confirm AGS; it must be interpreted alongside clinical history. Unfortunately, there is no correlation between sIgE concentration and symptom severity in AGS, an ambiguous situation also known from other forms of food allergy [32,33,34,35,36].

In our study, 22/33 α-Gal-sensitized patients could clearly be identified by ImmunoCAP (Table 1). An α-Gal-specific IgE binding could be confirmed by ICRIP in all these cases (Table 3). The remaining 11 patient sera that were sIgE-negative in ImmunoCAP (below 0.35 kU/L) were further analyzed by ICRIP: 6 patient sera clearly showed a positive signal for sIgE against Bos d TG (P01, P02, P03, P09, P18, and P29) and 1 serum against CTX (P13). Four sera were sIgE-negative in ImmunoCAP and ICRIP (P05, P07, P15, and P25). Of the latter patients, two sera (P07 and P15) did not show any signal in ImmunoCAP or ICRIP; thus, sIgE against α-Gal can be ruled out almost certainly as the cause for the clinical symptoms of these patients. An oral provocation test would be necessary to verify these negative results if the suspicion of meat allergy were still valid. Patient P05 showed positive sIgE binding in ImmunoCAP to total milk protein and Bos d 5 without an association with α-Gal. In the ICRIP analysis of the serum from patient P25, a weak sIgE signal could be detected for Bos d 6, while no sIgE signal for Bos d 6 could be detected in ImmunoCAP (<0.1 kU/L, Table 2). Interestingly, the ImmunoCAP analyses of three patient sera (P13, P21, and P29) showed weak sIgE binding to non-α-Gal-containing major milk allergens Bos d 4, Bos d 5, and Bos d 8, although a clearly positive sIgE signal above the threshold of 0.35 kU/L was only detectable in serum from patient P29 binding to Bos d 4 (Table 2).

The reason for a negative sIgE signal in ImmunoCAP and a positive sIgE signal in ICRIP (as observed only for P13) could be the antigen (α-Gal) density, the form of the antigen presentation (flexibility of the structure bound to the small carbohydrate moiety α-Gal), and the affinity of individual sIgE antibodies from patient sera to α-Gal. Unfortunately, there is no identifiable association of diagnostic results from ImmunoCAP or ICRIP for sIgE against α-Gal and specific clinical symptoms or phenotypes. It is important to note that CTX, IFX, and Bos d TG (bovine thyroglobulin) also contain the immunogenic glycans N-glycosyl neuraminic acid (Neu5Gc) and high mannose (Figure 2). The α-Gal and (non-human) sialic acid content of these analytes was previously verified by a lectin binding assay [27]. It remains to be determined if also other glycan epitopes, such as non-human N-glycolyl neuraminic acid, may play a role in the development of sIgE and glycan-associated forms of allergy [27,37]. SPT—like in vitro IgE-antibody detection—reveals the patient’s sensitization, and these methods are considered essential for diagnosis together with the clinical history. However, to date, no authorized SPT preparations for α-Gal are presently on the market, and therefore, the application of cetuximab for SPT must be considered “off-label-use”. Basophil activation test (BAT) may be useful in this context, however, as whole meat/innards extracts would have to be included, which are not standardized and may still contain non-specifically stimulating substances [38]. Recently, it was described that repeated α-Gal sensitization by tick bites over years is not associated with IgG4 antibodies, which is known from other types of allergy and desensitization procedures as well [22]. It remains to be determined if carbohydrate antigens elicit a different immune response than common protein allergens. The limitations of our work are the relatively small sample size (n = 33) and that not for all patients, oral food challenge data are available.

## 5. Conclusions

Seven patients with suspected AGS based on clinical symptoms benefitted from additional ICRIP analyses (P01, P02, P03, P09, P13, P18, and P29). Sera from these patients were negative in ImmunoCAP but showed a signal for α-Gal-specific IgE binding in the ICRIP diagnostic system. These analyses may support the diagnosis of AGS for these patients. Additionally, an association with IgE against α-Gal could be most likely ruled out for P05 (association with sIgE against milk protein and Bos d 5) and P25 (weak association with Bos d 6 in ICRIP, negative in ImmunoCAP). Taken together, this shows the benefit of the complementary use of the ICRIP diagnostic system in addition to ImmunoCAP for selected patients with suspected AGS.

## Figures and Tables

**Figure 1 nutrients-17-01541-f001:**
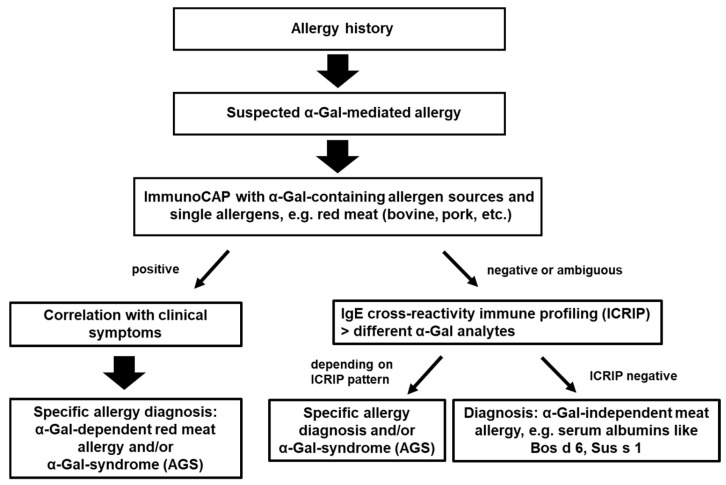
Diagnostic strategy for the detection of α-Gal-dependent and -independent food allergy.

**Figure 2 nutrients-17-01541-f002:**
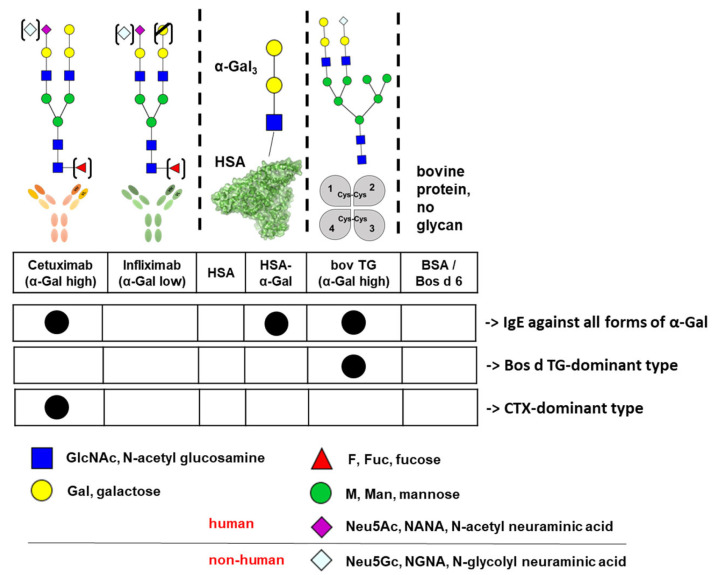
Prototypic ICRIP-binding patterns associated with the non-human, immunogenic glycan structures α-Gal and also Neu5Gc. Infliximab was used as a negative control antibody for IgE binding to CTX, HSA was a negative control for IgE against HSA-α-Gal, and BSA/Bos d 6 was a negative control for IgE-binding to (non-glycosylated) bovine proteins.

**Figure 3 nutrients-17-01541-f003:**
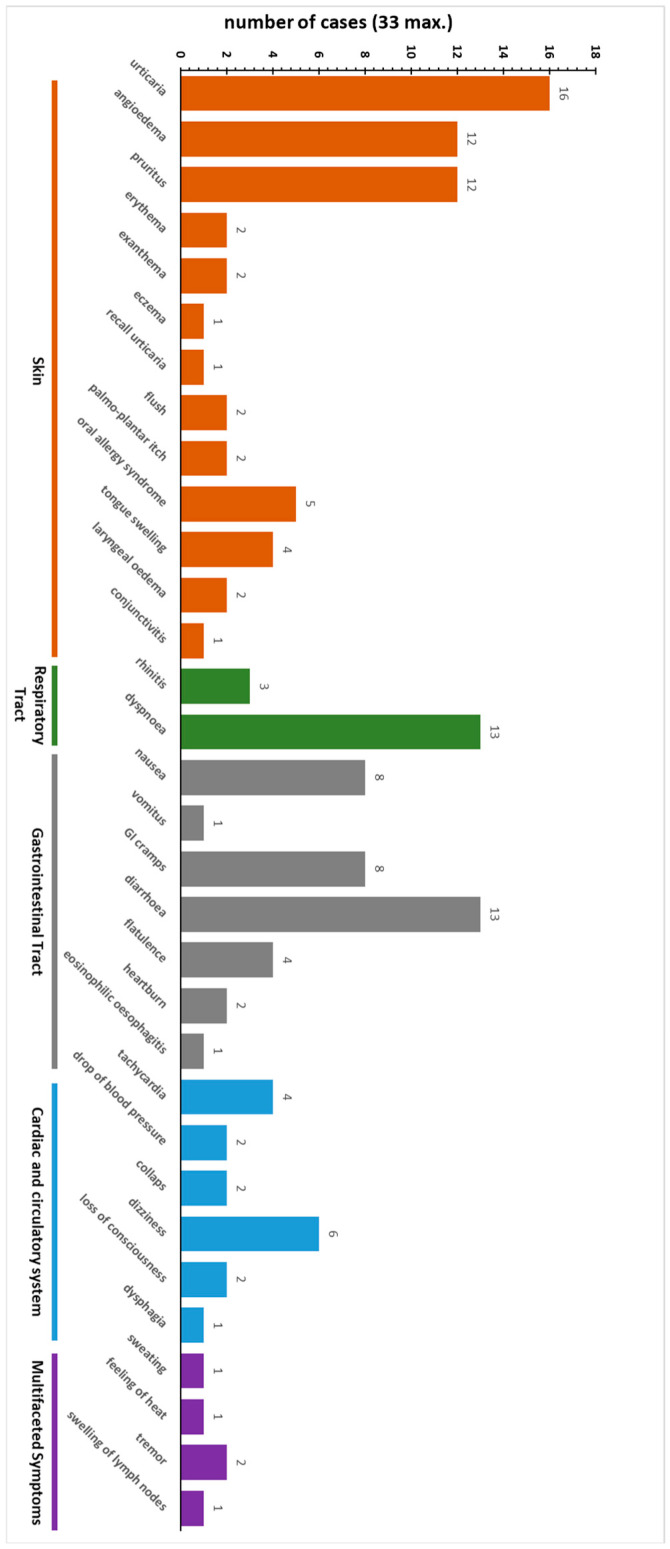
Clinical symptoms of patients with suspected AGS. Color code represents organ affection. In most patients, multiple organs were affected (for individual characteristics, see Appendix A).

**Figure 4 nutrients-17-01541-f004:**
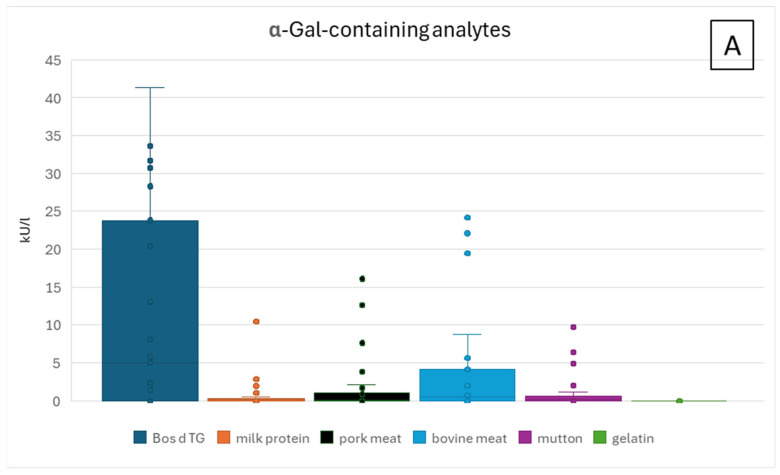
(**A**,**B**). ImmunoCAP results of patients with suspected AGS with analytes containing α-Gal (**A**) or not (**B**). For visualization purposes, y axis of (**A**) is cut off at 45 kU/L, and therefore, does not show 3 results of Bos d TG above this threshold (see Table 1). Cut off for data presentation in (**B**) was set at 1 kU/L. Y axis starts at 0 kU/L; however, the minimum threshold in ImmunoCAP for a positive sIgE signal is 0.1 kU/L.

**Table 1 nutrients-17-01541-t001:** ImmunoCAP results for patients with suspected α-Gal-associated food allergy performed with α-Gal-containing analytes. n.d, not determined due to lack of serum. Grey shading indicates negative results for IgE against Bos d TG, the commercial α-Gal-analyte. Values are provided as kU/L.

ID	Bos d TG	Milk Protein	Pork Meat	Bovine Meat	Mutton	Gelatin
**P01**	<0.1	<0.1	<0.1	0.27	<0.1	<0.1
**P02**	<0.1	<0.1	<0.1	<0.1	<0.1	<0.1
**P03**	<0.1	<0.1	<0.1	<0.1	<0.1	<0.1
**P04**	5.62	0.46	0.65	1.14	0.63	<0.1
**P05**	<0.1	0.17	<0.1	<0.1	<0.1	<0.1
**P06**	>100	0.31	2.15	5.65	0.61	<0.1
**P07**	0.11	<0.1	<0.1	<0.1	n.d.	<0.1
**P08**	28.3	0.21	<0.1	<0.1	<0.1	<0.1
**P09**	<0.1	<0.1	<0.1	<0.1	n.d.	<0.1
**P10**	4.96	<0.1	<0.1	0.14	<0.1	<0.1
**P11**	33.6	0.23	0.43	0.98	0.36	<0.1
**P12**	8.7	0.29	0.27	0.69	0.38	<0.1
**P13**	0.10	0.16	0.18	0.13	0.16	0.12
**P14**	30.7	2.83	16.1	22.1	6.39	<0.1
**P15**	<0.1	<0.1	<0.1	<0.1	<0.1	<0.1
**P16**	20.4	0.23	1.64	4.84	2.61	<0.1
**P17**	8.10	<0.1	0.86	2.03	0.17	<0.1
**P18**	<0.1	<0.1	<0.1	<0.1	<0.1	<0.1
**P19**	41.3	1.95	12.6	24.2	4.86	<0.1
**P20**	23.8	0.27	3.86	8.8	0.58	<0.1
**P21**	65.3	2.57	7.62	19.5	6.57	0.2
**P22**	31.7	<0.1	0.94	4.8	0.48	<0.1
**P23**	5.96	0.14	0.89	2.41	0.26	<0.1
**P24**	13.1	<0.1	1.26	4.18	1.99	<0.1
**P25**	<0.1	<0.1	<0.1	0.32	<0.1	<0.1
**P26**	1.61	<0.1	0.12	0.30	<0.1	<0.1
**P27**	1.41	0.20	0.1	0.55	<0.1	<0.1
**P28**	0.48	<0.1	<0.1	<0.1	<0.1	<0.1
**P29**	<0.1	1.0	<0.1	0.79	<0.1	<0.1
**P30**	>100	10.5	12.7	46.5	9.72	0.17
**P31**	1.94	<0.1	<0.1	0.23	<0.1	<0.1
**P32**	2.29	n.d.	0.12	0.44	n.d.	<0.1
**P33**	5.79	0.18	1.04	2.09	1.19	<0.1

**Table 2 nutrients-17-01541-t002:** ImmunoCAP results for patients with suspected α-Gal-associated food allergy performed with non- α-Gal-containing analytes. Grey shading indicates a negative result for IgE against Bos d TG, the commercial α-Gal analyte; see Table 1. Values are provided as kU/L.

ID	Bos d 4 (α-Lactalbumin)	Bos d 5 (β-Lactoglobulin)	Bos d 6 (BSA)	Bos d 8 (Casein)
**P01**	<0.1	<0.1	<0.1	<0.1
**P02**	<0.1	<0.1	<0.1	<0.1
**P03**	<0.1	<0.1	<0.1	<0.1
**P04**	<0.1	<0.1	<0.1	<0.1
**P05**	<0.1	0.13	<0.1	<0.1
**P06**	<0.1	<0.1	<0.1	<0.1
**P07**	<0.1	<0.1	<0.1	<0.1
**P08**	<0.1	<0.1	<0.1	<0.1
**P09**	<0.1	<0.1	<0.1	<0.1
**P10**	<0.1	<0.1	<0.1	<0.1
**P11**	<0.1	<0.1	<0.1	<0.1
**P12**	0.15	<0.1	<0.1	<0.1
**P13**	0.25	0.27	0.19	0.18
**P14**	<0.1	<0.1	<0.1	0.31
**P15**	<0.1	<0.1	<0.1	<0.1
**P16**	<0.1	<0.1	<0.1	<0.1
**P17**	<0.1	<0.1	<0.1	<0.1
**P18**	<0.1	<0.1	<0.1	<0.1
**P19**	<0.1	<0.1	<0.1	9.2
**P20**	<0.1	<0.1	<0.1	<0.1
**P21**	0.26	0.27	0.33	0.29
**P22**	<0.1	<0.1	<0.1	<0.1
**P23**	<0.1	<0.1	<0.1	<0.1
**P24**	<0.1	<0.1	<0.1	<0.1
**P25**	<0.1	<0.1	<0.1	<0.1
**P26**	<0.1	<0.1	<0.1	<0.1
**P27**	<0.1	<0.1	<0.1	<0.1
**P28**	<0.1	<0.1	<0.1	<0.1
**P29**	0.73	0.28	<0.1	0.35
**P30**	0.22	<0.1	<0.1	<0.1
**P31**	<0.1	<0.1	<0.1	<0.1
**P32**	<0.1	<0.1	<0.1	<0.1
**P33**	<0.1	<0.1	<0.1	<0.1

**Table 3 nutrients-17-01541-t003:** ICRIP assay results, patient ID in grey = CAP-negative for Bos d TG (commercial “α-Gal” analyte), positive ICRIP results shown as “+”; negative ICRIP results shown as “-”; n.d., not determined due to lack of serum.

ID	CTX	IFX	HSA	HSA-α-Gal_3_	Bos d TG	Bos d 6
**P01**	-	-	-	-	**+**	**n.d.**
**P02**	-	-	-	-	**+**	-
**P03**	-	-	-	-	**+**	-
**P04**	**+**	-	-	+	-	-
**P05**	-	-	-	-	-	-
**P06**	**+**	-	-	**+**	**+**	-
**P07**	-	-	-	-	-	-
**P08**	**+**	**n.d.**	-	+	+	-
**P09**	+	-	-	-	**+**	-
**P10**	**+**	**+**	-	-	-	-
**P11**	**+**	-	-	**+**	**+**	-
**P12**	**+**	-	-	+	+	-
**P13**	**+**	-	-	-	-	-
**P14**	**+**	-	-	**+**	**+**	-
**P15**	-	-	-	-	-	-
**P16**	**+**	-	-	**+**	**+**	-
**P17**	**+**	-	-	+	+	-
**P18**	-	-	-	-	**+**	-
**P19**	**+**	-	-	**+**	**+**	-
**P20**	**+**	-	-	-	-	-
**P21**	**+**	-	-	**+**	**+**	-
**P22**	**+**	-	-	**+**	**+**	-
**P23**	**+**	-	-	+	**+**	-
**P24**	**+**	-	-	-	-	-
**P25**	-	-	-	-	-	+
**P26**	**+**	-	-	-	-	-
**P27**	+	-	-	-	**+**	**+**
**P28**	+	-	-	-	-	-
**P29**	-	-	-	-	**+**	-
**P30**	**+**	**+**	-	**+**	**+**	-
**P31**	**+**	**-**	-	-	**+**	-
**P32**	**+**	-	-	-	-	-
**P33**	**+**	-	-	**+**	**+**	-

## Data Availability

Data are contained within the article and the Appendix A.

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
