# Peer review of "Connecting Diagnostics and Clinical Relevance of the α-Gal Syndrome—Individual Sensitization Patterns of Patients with Suspected α-Gal-Associated Allergy"

_nutrients, 2025, doi:10.3390/nu17091541_

Round 1
Reviewer 1 Report
Comments and Suggestions for Authors
Comments
In this paper, Jappe et al. explored diagnostic methods for α-Gal syndrome (AGS) using ImmunoCAP and the IgE cross-reactivity immune profiling (ICRIP) system, establishing an advanced diagnostic procedure. Specifically, the ICRIP analysis helped identify an additional seven patients with suspected AGS that were not detected by ImmunoCAP analysis alone. The combination of ImmunoCAP and ICRIP appears to offer significant advantages for the diagnosis of AGS, especially considering that the skin prick test is not yet commercially available. Overall, this is an interesting manuscript, but I have several concerns that need to be addressed before further consideration.
- Introduction has so much random information and some of it lacks references. Please update to simplify the text and add appropriate references.
2. Figure 1: It's better to have patient number (n=) in diagnostic strategy then easily follow patients along with authors later explanation.
- 2.2. Detection of allergen-specific IgE antibodies by ImmunoCAP, Line 135, Note that the cut-off commonly used for allergy diagnostics is 0.35 kU/l: Please explain in detail why a cut-off value of 0.35 kU/l was appropriate.
- The information on the incidence of AGS is completely missing. Please include the information.
- Have you considered patient geography? Do patients live in areas similar to where Lone Star ticks live?
- Figure 2: The page order was wrong. Figure 2 appeared after Figure 3. Please update the image location.
- Figure 3: As authors mention in the figure legend, I understand that Figure 3A and 3B, y-axis was cut off at 45 kU/l, however, in Figure 3B, comparing different allergens/different thresholds then it's not necessary to use the same cut-off in Figure 3B. I think it should be better to use much lower number of cut-off in Figure 3B, otherwise it won't be able to see the patient's values.
- Figure 4: There is no yellow square in the structures. Please double check the figure and update the picture and legend accordingly.
Specific comments
- Line 45-55: The explanation is still so many unknown factors. It should be better to have many strong references to support your explanation.
- Line 54: “classical””, this must be “classical”. Please remove the extra double quotation mark.
- Line 192: 3.1.2..,please remove the extra dot.
- Some units of kU/l were kU/L. Please format the unit accordingly and use the same format consistently.
Author Response
We, the authors, want to express our thanks for the reviewers` comments which were very helpful und have been worked upon thoroughly in order to further improve our manuscript
Comments of Reviewer 1
“……Overall, this is an interesting manuscript, but I have several concerns that need to be addressed before further consideration.”
- Introduction has so much random information and some of it lacks references. Please update to simplify the text and add appropriate references.
Thank you for the suggestion! Introduction was revised, references were added as indicated in red in the revised submitted manuscript. However, we aimed intentionally at providing the description of the particularities of the alpha-Gal syndrome because the scope of the journal is not primarily allergy but nutritional medicine.
- Figure 1: It's better to have patient number (n=) in diagnostic strategy then easily follow patients along with authors later explanation.
Figure 1 shall represent a decision tree for AGS diagnosis in general and is not meant to be only connected to the patients investigated in the presented study, which is why no patient numbers referring to this specific manuscript were included. We, therefore, prefer to leave the patient numbers out.
- 2.2. Detection of allergen-specific IgE antibodies by ImmunoCAP, Line 135. Note that the cut-off commonly used for allergy diagnostics is 0.35 kU/l: Please explain in detail why a cut-off value of 0.35 kU/l was appropriate.
Thank you for this important comment! In routine allergy diagnostic tests, a cut-off value of 0.35 kU/l is considered to represent clinical relevance. It was also described in other publications concerning specific IgE against alpha-Gal, so for example, Lesmana et al., observed differences in the AGS dynamics (symptom onset) between patients with sIgE in the range of 0.35-0.7 kU/l when compared to higher concentrations 0.7-3.5 kU/l) which was added to the text (lines 145-149) together with the respective references were added as appropriate: Lesmana et al., German-Sanchez et al.
- The information on the incidence of AGS is completely missing. Please include the information.
Thank you for this important comment! Information on incidence and also prevalence has now been added to the introduction, lines 68-74.
- Have you considered patient geography? Do patients live in areas similar to where Lone Star ticks live?
The lone star tick, which is most likely responsible for a part of the patient sensitization to alpha-Gal in the US, is not endemic in Germany. So, the answer is no. However, other tick species are known to be responsible for alpha-Gal sensitization in Germany and other European countries which we have briefly pointed out (lines 53-54).
- Figure 2: The page order was wrong. Figure 2 appeared after Figure 3. Please update the image location.
Figures were re-organized and renamed accordingly throughout the manuscript.
- Figure 3: As authors mention in the figure legend, I understand that Figure 3A and 3B, y-axis was cut off at 45 kU/l, however, in Figure 3B, comparing different allergens/different thresholds then it's not necessary to use the same cut-off in Figure 3B. I think it should be better to use much lower number of cut-off in Figure 3B, otherwise it won't be able to see the patient's values.
Now Figure 4: Same cut-off was used for both graphs to ensure comparability. However, we see the point of the reviewer and changed the cut-off for the presentation of the results in Figure 4B to 1 kU/l in order to resolve the difference in these low IgE concentrations.
- Figure 4: There is no yellow square in the structures. Please double check the figure and update the picture and legend accordingly.
Corrected accordingly, please note: Figure was renamed and inserted in the Methods section.
Specific comments
- Line 45-55: The explanation is still so many unknown factors. It should be better to have many strong references to support your explanation.
Thank you for this valuable comment! Citations were added for these statements.
- Line 54: “classical””, this must be “classical”. Please remove the extra double quotation mark.
Corrected accordingly.
- Line 192: 3.1.2..,please remove the extra dot.
Corrected accordingly.
- Some units of kU/l were kU/L. Please format the unit accordingly and use the same format consistently.
Corrected accordingly throughout the manuscript.
Comments and Suggestions for Authors by Reviewer 2
“….This component-resolved diagnostic approach, integrating clinical history with laboratory testing, significantly improves AGS diagnostic accuracy, enables personalized management, and effectively prevents accidental exposure and symptom recurrence - thereby addressing the current clinical problem of AGS underdiagnosis. The study establishes that combined ImmunoCAP and ICRIP testing represents an optimized new standard for AGS diagnosis. The research makes three key contributions: (1) development of the novel ICRIP detection system, (2) establishment of a combined diagnostic strategy, and (3) resolution of limitations inherent to current standalone ImmunoCAP testing. This work provides an important new methodology for improving diagnosis of this unique food allergy syndrome, with significant clinical value. “
We wish to thank the reviewer for his positive comment on our study.
“Given these advancements, I recommend acceptance after addressing critical revisions.
- Abstract Section: Lines 16–17: The phrase "the diagnosis is challenging" is too vague. Suggest specifying the challenges, e.g., "due to delayed symptom onset and cross-reactivity with multiple mammalian products."
Corrected accordingly, thank you for the comment!
- Introduction Section: Lines 44–45: The statement "Only for infliximab...observed so far" needs updated references. Recommend citing post-2020 studies on other biologics linked to α-Gal.
Thank you for the valuable comment! This part of the introduction was rephrased and citations were added (lines 45-47).
- Methods Section: Tables 1 & 2: Footnotes are insufficient. Clarify units ("values in kU/L") and the meaning of grey highlighting ("grey shading indicates...").
Thank you for your comment! A note referring to units was provided, and the explanation for “grey highlighting” was revised accordingly.
- Lines 110–111: The ethics statement should specify how informed consent was obtained (e.g., written, verbal).
The patients gave informed consent in writing, which has now been described in the manuscript (lines 120-121). (In 4 patients, two of these “cases” published as case reports already and cited accordingly, have primarily been treated in another department. I trustingly assume that the signed forms are filed there.)
- Results Section: Figure 2 (clinical symptom distribution) should be described in detail in the text (currently only briefly mentioned in lines 178–179).
Thank you for this comment! Symptoms and symptom distribution as well as patient groups referred to in Figure 3 (former Figure 2) have been further described in lines 186-202. We hope the reviewer agrees with this part.
- Lines 213–215: The case of ID 1873 requires statistical significance analysis to support claims.
Thank you for this comment! The authors would like to make the point, that the situation of patient ID 1873, now documented throughout the manuscript as P33, was included in the manuscript as a description of a specific patient situation with no claim of significance of the measured IgE levels. However, to clarify the patient description for the reader, we shifted the paragraph to the Discussion section, lines 325-329.
- Discussion Section: Lines 272–273: The claim that AGS is "underreported" needs recent epidemiological data for support.
Thank you for this comment! In my clinical practice I am confronted with patients having travelled far to seek my advice because their medical doctors “do not believe, this meat allergy exists”. The delay in symptom development, and the lack of knowledge on this entity in medical personnel most certainly causes non diagnosis and underreporting, which is why I refrained from adding epidemiological data for support: Those investigators performing epidemiological studies KNOW what they are looking for, which does not seem to be the case in more general clinical practice. Actually, Mollah and co-authors most recently concluded from their epidemiology study an underreporting. We added explanatory words to this effect to the text, lines 298-301.
- Lines 316–317: The statement that Ig E levels do not correlate with symptom severity should include possible mechanistic explanations.
Thank you very much for this comment! The situation has been clarified further and references were added (lines 342-345). However, we refrained from adding the different mechanistic hypotheses here, like the fact that alpha-Gal is both, associated with lipids as well as with proteins, aspects that may account for the differences in symptom development dynamics and their severity. It is very interesting but complex as well and might strain the format of this study. The authors hope that the reviewer is nonetheless satisfied with the revision.
- Figures & Tables: Figure 4 (ICRIP analysis schematic) should be introduced earlier in the Methods section.
Figure 4 was inserted into the ICRIP Methods section 2.3 and renamed as Figure 2. All Figures were renamed accordingly.
- Supplementary Tables S1–S3 should be referenced and analyzed more thoroughly in the main text.
Thank you for bringing the attention to the proper embedding of the supplementary material! Reference to the supplementary material was provided in the manuscript as follows: S1 and S2 were referred to in Methods Section “study group”, lines 115 and 121 and, Methods 2.2., line 142, as well as Results section 3.1., line 194. Table S1 was added to Results section 3.1.1, lines 203-204, references to Table S2, Table S3 were added to Results Section 3.1.4, line 274, Table S1 was discussed in lines 293-296 in the Discussion section.
- Terminology Consistency: The manuscript alternates between "α-Gal" and "a-Gal"—standardize to "α-Gal" throughout.
Checked and corrected throughout the manuscript.
- Clinical Relevance: Lines 80–82: The discussion on clinical judgment should include a decision tree or flowchart for practical guidance.
Thank you for this comment! The authors` aim was to provide a practical guidance by flow scheme/decision tree as described in Figure 1 with particular reference to ICRIP. A clinical algorithm including a note to the multi-analyte array described here has recently been published under participation of the first author of the present study (Darsow,…Jappe,… et al., 2024) which is why we focussed particularly on ICRIP in our flowchart here.
- Highlighting Innovation: Lines 146–147: The novelty of the ICRIP method should be explicitly compared to existing techniques (e.g., ImmunoCAP).
Thank you very much for this comment! The overall focus of this work was the comparison and the mutual complementation of these two analysis methods. The authors are of the opinion that the ImmunoCAP system will have its use in alpha-Gal diagnostics as a routine tool, but that the tailored ICRIP system will help physicians/allergologists in doubtful situation and add diagnostic value. In our work we could show that additional patients could be identified by adding ICRIP to the ImmunoCAP routine tests.
- Study Limitations: Add a dedicated paragraph on limitations, such as small sample size (n=33) and lack of oral food challenge data.
Thank you for the comment! The information about study limitation was added at the end of the Discussion section (lines 386-388).
Additional changes
We changed the ID numbers to patient numbers in order to ensure additional data protection. The changes are also highlighted in red.

Reviewer 2 Report
Comments and Suggestions for Authors
This study addresses the diagnostic challenges of α-Gal syndrome (AGS) by developing a novel combined testing strategy. The authors analyzed sera from 33 suspected AGS patients using both conventional Immuno CAP assays and their newly developed Ig E cross-reactivity immune profiling (ICRIP) system. Key findings demonstrate: Immuno CAP identified 22 positive cases (>0.35 kU/L), while ICRIP detected an additional 7 α-Gal sensitized cases among the remaining 11 negative/equivocal samples, increasing the total diagnostic yield from 66.7% to 87.9%. This component-resolved diagnostic approach, integrating clinical history with laboratory testing, significantly improves AGS diagnostic accuracy, enables personalized management, and effectively prevents accidental exposure and symptom recurrence - thereby addressing the current clinical problem of AGS underdiagnosis. The study establishes that combined Immuno CAP and ICRIP testing represents an optimized new standard for AGS diagnosis.
The research makes three key contributions: (1) development of the novel ICRIP detection system, (2) establishment of a combined diagnostic strategy, and (3) resolution of limitations inherent to current standalone Immuno CAP testing. This work provides an important new methodology for improving diagnosis of this unique food allergy syndrome, with significant clinical value. Given these advancements, I recommend acceptance after addressing critical revisions.
- Abstract Section : Lines 16–17: The phrase "the diagnosis is challenging" is too vague. Suggest specifying the challenges, e.g., "due to delayed symptom onset and cross-reactivity with multiple mammalian products."
- Introduction Section : Lines 44–45: The statement "Only for infliximab...observed so far" needs updated references. Recommend citing post-2020 studies on other biologics linked to α-Gal.
- Methods Section : Tables 1 & 2: Footnotes are insufficient. Clarify units ("values in kU/L") and the meaning of grey highlighting ("grey shading indicates...").
- Lines 110–111: The ethics statement should specify how informed consent was obtained (e.g., written, verbal).
- Results Section :Figure 2 (clinical symptom distribution) should be described in detail in the text (currently only briefly mentioned in lines 178–179).
- Lines 213–215: The case of ID 1873 requires statistical significance analysis to support claims.
- Discussion Section :Lines 272–273: The claim that AGS is "underreported" needs recent epidemiological data for support.
- Lines 316–317: The statement that Ig E levels do not correlate with symptom severity should include possible mechanistic explanations.
- Figures & Tables: Figure 4 (ICRIP analysis schematic) should be introduced earlier in the Methods section.
- Supplementary Tables S1–S3 should be referenced and analyzed more thoroughly in the main text.
- Terminology Consistency: The manuscript alternates between "α-Gal" and "a-Gal"—standardize to "α-Gal" throughout.
- Clinical Relevance: Lines 80–82: The discussion on clinical judgment should include a decision tree or flowchart for practical guidance.
- Highlighting Innovation: Lines 146–147: The novelty of the ICRIP method should be explicitly compared to existing techniques (e.g., ImmunoCAP).
- Study Limitations: Add a dedicated paragraph on limitations, such as small sample size (n=33) and lack of oral food challenge data.
Author Response
Comments and Suggestions for Authors by Reviewer 2
“….This component-resolved diagnostic approach, integrating clinical history with laboratory testing, significantly improves AGS diagnostic accuracy, enables personalized management, and effectively prevents accidental exposure and symptom recurrence - thereby addressing the current clinical problem of AGS underdiagnosis. The study establishes that combined ImmunoCAP and ICRIP testing represents an optimized new standard for AGS diagnosis. The research makes three key contributions: (1) development of the novel ICRIP detection system, (2) establishment of a combined diagnostic strategy, and (3) resolution of limitations inherent to current standalone ImmunoCAP testing. This work provides an important new methodology for improving diagnosis of this unique food allergy syndrome, with significant clinical value. “
We wish to thank the reviewer for his positive comment on our study.
“Given these advancements, I recommend acceptance after addressing critical revisions.
- Abstract Section: Lines 16–17: The phrase "the diagnosis is challenging" is too vague. Suggest specifying the challenges, e.g., "due to delayed symptom onset and cross-reactivity with multiple mammalian products."
Corrected accordingly, thank you for the comment!
- Introduction Section: Lines 44–45: The statement "Only for infliximab...observed so far" needs updated references. Recommend citing post-2020 studies on other biologics linked to α-Gal.
Thank you for the valuable comment! This part of the introduction was rephrased and citations were added (lines 45-47).
- Methods Section: Tables 1 & 2: Footnotes are insufficient. Clarify units ("values in kU/L") and the meaning of grey highlighting ("grey shading indicates...").
Thank you for your comment! A note referring to units was provided, and the explanation for “grey highlighting” was revised accordingly.
- Lines 110–111: The ethics statement should specify how informed consent was obtained (e.g., written, verbal).
The patients gave informed consent in writing, which has now been described in the manuscript (lines 120-121). (In 4 patients, two of these “cases” published as case reports already and cited accordingly, have primarily been treated in another department. I trustingly assume that the signed forms are filed there.)
- Results Section: Figure 2 (clinical symptom distribution) should be described in detail in the text (currently only briefly mentioned in lines 178–179).
Thank you for this comment! Symptoms and symptom distribution as well as patient groups referred to in Figure 3 (former Figure 2) have been further described in lines 186-202. We hope the reviewer agrees with this part.
- Lines 213–215: The case of ID 1873 requires statistical significance analysis to support claims.
Thank you for this comment! The authors would like to make the point, that the situation of patient ID 1873, now documented throughout the manuscript as P33, was included in the manuscript as a description of a specific patient situation with no claim of significance of the measured IgE levels. However, to clarify the patient description for the reader, we shifted the paragraph to the Discussion section, lines 325-329.
- Discussion Section: Lines 272–273: The claim that AGS is "underreported" needs recent epidemiological data for support.
Thank you for this comment! In my clinical practice I am confronted with patients having travelled far to seek my advice because their medical doctors “do not believe, this meat allergy exists”. The delay in symptom development, and the lack of knowledge on this entity in medical personnel most certainly causes non diagnosis and underreporting, which is why I refrained from adding epidemiological data for support: Those investigators performing epidemiological studies KNOW what they are looking for, which does not seem to be the case in more general clinical practice. Actually, Mollah and co-authors most recently concluded from their epidemiology study an underreporting. We added explanatory words to this effect to the text, lines 298-301.
- Lines 316–317: The statement that Ig E levels do not correlate with symptom severity should include possible mechanistic explanations.
Thank you very much for this comment! The situation has been clarified further and references were added (lines 342-345). However, we refrained from adding the different mechanistic hypotheses here, like the fact that alpha-Gal is both, associated with lipids as well as with proteins, aspects that may account for the differences in symptom development dynamics and their severity. It is very interesting but complex as well and might strain the format of this study. The authors hope that the reviewer is nonetheless satisfied with the revision.
- Figures & Tables: Figure 4 (ICRIP analysis schematic) should be introduced earlier in the Methods section.
Figure 4 was inserted into the ICRIP Methods section 2.3 and renamed as Figure 2. All Figures were renamed accordingly.
- Supplementary Tables S1–S3 should be referenced and analyzed more thoroughly in the main text.
Thank you for bringing the attention to the proper embedding of the supplementary material! Reference to the supplementary material was provided in the manuscript as follows: S1 and S2 were referred to in Methods Section “study group”, lines 115 and 121 and, Methods 2.2., line 142, as well as Results section 3.1., line 194. Table S1 was added to Results section 3.1.1, lines 203-204, references to Table S2, Table S3 were added to Results Section 3.1.4, line 274, Table S1 was discussed in lines 293-296 in the Discussion section.
- Terminology Consistency: The manuscript alternates between "α-Gal" and "a-Gal"—standardize to "α-Gal" throughout.
Checked and corrected throughout the manuscript.
- Clinical Relevance: Lines 80–82: The discussion on clinical judgment should include a decision tree or flowchart for practical guidance.
Thank you for this comment! The authors` aim was to provide a practical guidance by flow scheme/decision tree as described in Figure 1 with particular reference to ICRIP. A clinical algorithm including a note to the multi-analyte array described here has recently been published under participation of the first author of the present study (Darsow,…Jappe,… et al., 2024) which is why we focussed particularly on ICRIP in our flowchart here.
- Highlighting Innovation: Lines 146–147: The novelty of the ICRIP method should be explicitly compared to existing techniques (e.g., ImmunoCAP).
Thank you very much for this comment! The overall focus of this work was the comparison and the mutual complementation of these two analysis methods. The authors are of the opinion that the ImmunoCAP system will have its use in alpha-Gal diagnostics as a routine tool, but that the tailored ICRIP system will help physicians/allergologists in doubtful situation and add diagnostic value. In our work we could show that additional patients could be identified by adding ICRIP to the ImmunoCAP routine tests.
- Study Limitations: Add a dedicated paragraph on limitations, such as small sample size (n=33) and lack of oral food challenge data.
Thank you for the comment! The information about study limitation was added at the end of the Discussion section (lines 386-388).
Additional changes
We changed the ID numbers to patient numbers in order to ensure additional data protection. The changes are also highlighted in red.

Round 2
Reviewer 2 Report
Comments and Suggestions for Authors
accept